# A Comprehensive Investigation of the Properties of a Five-Phase Induction Motor Operating in Hazardous States in Various Connections of Stator Windings

Jakub Kellner *, Slavomír Kaščák *, Michal Praženica 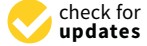 and Patrik Resutík

Department of Mechatronics and Electronics, Faculty of Electrical Engineering and Information Technology, University of Zilina, 010 01 Žilina, Slovakia; michal.prazenica@feit.uniza.sk (M.P.); patrik.resutik@feit.uniza.sk (P.R.)
* Correspondence: jakub.kellner@feit.uniza.sk (J.K.); slavomir.kascak@feit.uniza.sk (S.K.)

**Abstract:** This paper examines the properties of a multi-phase drive for EV (electric vehicles) and HEV (hybrid-electric vehicles) using a simulation model in the Matlab/Simulink environment and verifies the findings by experimental measurements on a real motor. The paper studies a five-phase induction motor, a suitable alternative for electric vehicles, due to its better properties such as better torque, smoother ripple, better fault tolerance, and the possibility of connecting stator windings to star, pentagon, and pentagram. The fundamentals of the article are to find out how this engine behaves in fault states, which can be called hazardous states. The paper presents a comprehensive evaluation of the decrease of mechanical power, torque, and power losses during motor operation without failure, in case of failure of one phase, and in case of failure of two adjacent phases and two non-adjacent phases, in different connections. In the simulations, the five-phase drive is powered from an ideal five-phase voltage source to verify the behavior of losses on the motor in fault conditions. Subsequently, the motor model is powered by a five-phase VSI, while the simulated waveforms are confirmed on a real motor, which is also powered by a five-phase VSI. The investigation results are the detection, which of the stator windings has better properties in the fault-free state and the case of fault states in operation. For which stator windings connection, it is most advantageous to design and dimension a five-phase induction motor.

**Keywords:** five-phase induction motor; hazardous state; fault; mechanical power and torque characteristics; power losses; stator winding connection

## 1. Introduction

Three-phase electric motors have been used for a very long time for drives where variable speed is required. The main reason why they are often used is that the three-phase power supply is very easily available. Therefore, there is no problem with using three-phase motors. However, this is irrelevant when using electric motors for EV and HEV because in this case, power from inverters is needed, where DC power is converted to AC, most often by VSI or matrix converters, which are not popular, but for their complex control. Thus, this shortcoming is eliminated, and the number of phases for powering the electric motor can be arbitrary. Depending on the drive requirements, an inverter with the required number of phases is created.

The use of a multi-phase electric motor has several advantages over a three-phase one. The main advantage is the reduction of the required power per inverter phase. It reduces the current without increasing the voltage and improves the inverter mechanical and electrical properties. Other benefits include improved efficiency by lowering stator winding losses, lower content of higher harmonics, reducing machine noise, increasing torque, and minimizing ripple [1–4].

Multi-phase electric motors have another feature against three-phase motors. These are the other degrees of freedom that these engines provide. These degrees of freedom can be used to reduce system failure. Failure of a single phase in a multi-phase machine will result in minimal power reduction, and the machine will still be able to operate automatically. A phase failure can occur on the power supply side, where the inverter branch switches can be destroyed. Failure of the conductive path connection between the inverter and the motor. Or on the motor side, where one motor winding can be destroyed or a failure at the motor connection terminals. Fault conditions and behavior of multi-phase motors during these operating conditions is a very important part of drive research.

The fundamentals of this article are to find out how the motor will behave in the event of a phase failure—detection of a decrease in the torque and power of the machine and a change in the electrical losses of the drive in the event of a loss of motor phase compared to a fault-free state. The error conditions for which the above properties have been examined are [5,6]:

- Failure of one phase
- Failure of two adjacent, consecutive phases
- Failure of two non-adjacent phases

Currently, the most frequently investigated multi-phase motors are 5-phase and 6-phase symmetrical and 6-phase unbalanced induction motors. If there is a loss of phase, an increase in motor losses is expected, which will increase machine warming, and that could have a destructive effect on the entire drive. Therefore, it is necessary to know the behavior of the machine during phase failure operation if it is not possible to switch off the drive.

We know that we can connect the stator windings to three different connections in a five-phase induction motor, each connection having other properties. It is a connection to the star, pentagon, and pentacle. Therefore, part of this study is to determine the properties of a five-phase induction motor in phase failure for all three stator winding connections. Research on fault conditions in a five-phase induction motor is reported in the literature [7–9]. The result of the analysis is that in case of failure of one phase, the most suitable solution is the connection to the pentagon. With this connection, the induction motor can operate with reduced power of 10%. When connected to a star, it is 20%. When connecting to the pentacle, the authors mention that this connection is not suitable for operation in the event of a failure of one phase.

The paper [10–12] presents simulations of a five-phase induction motor connected to a pentagon and a pentacle with the motor running without a single phase. These posts state that the engine loses its original power but can work without problems.

Several papers in the literature directly address a fault-tolerant control technique that minimizes motor winding losses in the event of a phase failure. In [13], the solution is realized using a phasor representation of each stator current in a steady state. Or [14,15], where a method based on a multi-space vector representation is used, and the transition to four-phase operation. However, these contributions do not report the situation in a phase failure event without a change in management—identifying which five-phase induction motor connection provides the best properties in terms of energy efficiency, power, or torque in a phase failure event. In [16], the theory around a five-phase machine model in the post-failure state is comprehensively solved. The article deals with obtaining a model from Clarke transformation. In recent years, articles [17–21] have appeared most frequently in the literature, which provides analysis and solution of fault tolerance in synchronous five-phase motors with permanent magnets.

This article provides a new view of engine behavior during fault conditions. The paper comprehensively evaluates the properties of a five-phase motor in the event of a phase failure in all three types of connection. It evaluates which type of connection of the stator windings has better properties in the fault-free and fault state. The result of this research is to find out what changes in torque, power, and losses occurring in fault conditions in different types of connections and which type of connection, star, pentagon, or pentacle connections have the best properties in the fault state.

## 2. Theory of Five-Phase Induction Motor

The five-phase induction motor works on the same principle as the three-phase. In both cases, Faraday laws and the Lorentz force acting on the conductor apply. A five-phase AC voltage is applied to the stator, which is shifted by 72° in space and time. The stator winding of an n-phase machine can be designed so that the spatial displacement of two adjacent phases is always:

$$\alpha = \frac{2\pi}{n} \tag{1}$$

In this way, a symmetrical multi-phase machine is created. Figure 1 shows the spatial distribution of the windings of a five-phase induction motor.

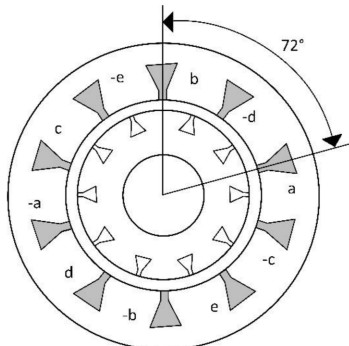

**Figure 1.** Winding distribution of a five-phase induction motor.

As mentioned above, in a five-phase machine, we distinguish three connections of the stator winding. Each of these connections has different properties. It is a star connection, pentagon, and pentacle connection. A schematic illustration of the connection is shown in Figure 2. The letters *a–e* in Figure 2 represent the power connection from a five-phase inverter. $W_A$–$W_E$ represent the individual phases of stator winding [22].

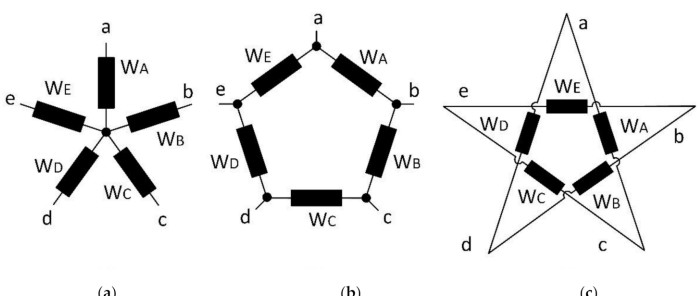

**Figure 2.** Five-phase motor stator winding connection configuration: (**a**) Star connection; (**b**) Pentagon connection; (**c**) Pentacle connection.

The difference between the individual connections is how the windings are interconnected. In the star connection, the voltage and thus the current on the motor winding is also equal to the phase voltage of the source. However, a different situation is in the pentagon connection. In this case, the resulting voltage on the stator windings is the difference between two adjacent phase voltages of the source, as shown in Figure 3a. It shows a phase diagram for a five-phase source connected to a pentagon. From the phase diagram, we see that the amplitude, in this case, is *1.1756* × $V_{ph}$. In the pentacle connection, the resulting voltage on the stator windings is the difference between two non-adjacent phase voltages of the source, as shown in Figure 3b, which shows a phase diagram for a pentacle connection. The voltage amplitude, in this case, is higher by *1.902* × $V_{ph}$, and this brings higher motor torques without changing the input voltage [23–25].

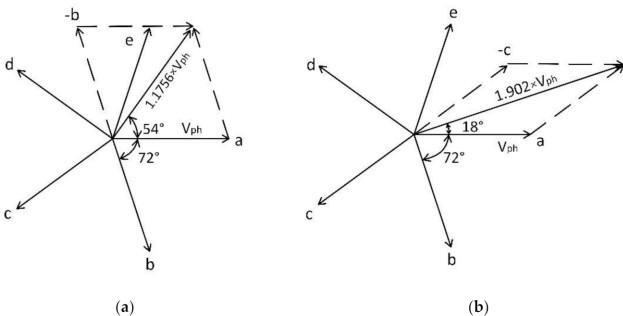

**Figure 3.** Phase diagram for five-phase source: (**a**) Mains voltage for adjacent phases (pentagon); (**b**) Mains voltage for non-adjacent phases (pentacle).

If the multi-phase machine is symmetrical, the motor power input is given as follows:

$$P_1 = m_1 \times U_1 \times I_1 \times \cos \varphi \ (W) \tag{2}$$

where:

- $m_1$ is the number of stator winding phases (-)
- $U_1$ is phase voltage (V)
- $I_1$ is phase current (A)
- $\cos\varphi$ is power factor (-)

The relation for calculation of mechanical torque:

$$T_{mech} = \frac{m_1 \times p \times U_1^2 \times \frac{R_2'}{s}}{2 \times \pi \times f_1 \times \left[ \left( R_1 + \frac{R_2'}{s} \right)^2 + \left( X_{r1} + X_{r20}' \right)^2 \right]} \ (Nm) \tag{3}$$

where:

- $R_2'/s$ is the total active rotor resistance ($\Omega$)
- $X_{r20}$ is leakage reactance of the rotor winding converted to a stator ($\Omega$)
- $X_{r1}$ is stator leakage reactance ($\Omega$)
- $R_1$ is the stator winding resistance ($\Omega$)
- $R_2'$ is the resistance of rotor converted to stator ($\Omega$)
- $p$ is the number of pole pairs (-)
- $s$ is the motor slip (-)

From Equation (3), we see that the mechanical torque is directly proportional to the square of the phase voltage of the winding. Therefore, it should be noted that when the stator windings of a five-phase induction motor are connected to a pentagon, the mechanical torque of the machine is increased by *1.382* times compared to the torque generated in the star connection, and when connected to a pentagram, it is up to *3.6176* times.

We know that if the motor is operating in fault-free operation, the stator winding currents are symmetrical. In this case, we can use the relation to calculate the power input given in Equation (2). However, a different situation occurs if a fault condition occurs, a phase failure. Then the currents are not symmetrical, and the power calculation must be calculated for each phase separately. However, the situation is different when calculating the power on the shaft. The power calculation on the motor shaft is as follows:

$$P = T \times \Omega = T \frac{2\pi n}{60} \ (W) \tag{4}$$

where:

- *T* is the motor torque (Nm)
- $\Omega$ is the angular velocity of the rotor (rad $\times$ s$^{-1}$)
- *n* is the motor speed (rpm)

In this article, the simulations and measurements will be presented, where the torque on the shaft is investigated, as well as the efficiency, the decrease of the power on the shaft, and the power taken from the supply system.

## 3. Modeling of Five-Phase Induction Motor

A substantial part of the work is based on examining the properties of a five-phase motor, so we present a mathematical model of the motor, which was used to perform simulations to verify the behavior of the motor in fault conditions. The principle of creating a mathematical model of a five-phase machine is the same as that of a three-phase motor. Variables such as voltage, current, or flux are transmitted to a reference frame. The voltage of the balanced five-phase induction machine is expressed as:

$$U_a = \sqrt{2}U_{rms}\sin(\omega t) \tag{5}$$

$$U_b = \sqrt{2}U_{rms}\sin\left(\omega t - \frac{2\pi}{5}\right) \tag{6}$$

$$U_c = \sqrt{2}U_{rms}\sin\left(\omega t - \frac{4\pi}{5}\right) \tag{7}$$

$$U_d = \sqrt{2}U_{rms}\sin\left(\omega t + \frac{4\pi}{5}\right) \tag{8}$$

$$U_e = \sqrt{2}U_{rms}\sin\left(\omega t + \frac{2\pi}{5}\right) \tag{9}$$

The transformation equation is expressed as follows [26–29]:

$$
\begin{bmatrix} U_q \\ U_d \\ U_x \\ U_y \\ U_0 \end{bmatrix} = \frac{2}{5}
\begin{bmatrix}
1 & \cos\left(\frac{2\pi}{5}\right) & \cos\left(\frac{4\pi}{5}\right) & \cos\left(\frac{6\pi}{5}\right) & \cos\left(\frac{8\pi}{5}\right) \\
0 & -\sin\left(\frac{2\pi}{5}\right) & -\sin\left(\frac{4\pi}{5}\right) & -\sin\left(\frac{6\pi}{5}\right) & -\sin\left(\frac{8\pi}{5}\right) \\
1 & \cos\left(\frac{6\pi}{5}\right) & \cos\left(\frac{12\pi}{5}\right) & \cos\left(\frac{18\pi}{5}\right) & \cos\left(\frac{24\pi}{5}\right) \\
0 & -\sin\left(\frac{6\pi}{5}\right) & -\sin\left(\frac{12\pi}{5}\right) & -\sin\left(\frac{18\pi}{5}\right) & -\sin\left(\frac{24\pi}{5}\right) \\
\frac{1}{2} & \frac{1}{2} & \frac{1}{2} & \frac{1}{2} & \frac{1}{2}
\end{bmatrix}
\begin{bmatrix} U_a \\ U_b \\ U_c \\ U_d \\ U_e \end{bmatrix} \tag{10}
$$

Stator voltage equation in the reference *dq* frame with rotating angular velocity *ωa* is:

$$U_{ds} = R_s i_{ds} - \omega_a \varphi_{qs} + p\varphi_{ds} \tag{11}$$

$$U_{qs} = R_s i_{qs} - \omega_a \varphi_{ds} + p\varphi_{qs}, \tag{12}$$

$$U_{xs} = R_s i_{xs} + p\varphi_{xs} \tag{13}$$

$$U_{ys} = R_s i_{ys} + p\varphi_{ys}, \tag{14}$$

$$U_{0s} = R_s i_{0s} + p\varphi_{0s}. \tag{15}$$

Rotor voltage equation:

$$U_{dr} = R_r i_{dr} + (\omega_a - \omega)\varphi_{qr} + p\varphi_{dr}, \tag{16}$$

$$U_{qr} = R_r i_{qr} + (\omega_a - \omega)\varphi_{dr} + p\varphi_{qr}, \tag{17}$$

$$U_{xr} = R_r i_{xr} + p\varphi_{xr} \tag{18}$$

$$U_{yr} = R_r i_{yr} + p\varphi_{yr}, \tag{19}$$

$$U_{0r} = R_r i_{0r} + p\varphi_{0r}. \tag{20}$$

Flux linkages of stator:

$$\varphi_{ds} = (L_{ls} + L_m)i_{qs} + L_m i_{qr}, \tag{21}$$

$$\varphi_{qs} = (L_{ls} + L_m)i_{ds} + L_m i_{dr}, \tag{22}$$

$$\varphi_{xs} = L_{ls} i_{xs} \tag{23}$$

$$\varphi_{ys} = L_{ls} i_{ys}, \tag{24}$$

$$\varphi_{0s} = L_{ls} i_{0s}. \tag{25}$$

Flux linkages of rotor:

$$\varphi_{dr} = (L_{lr} + L_m)i_{qr} + L_m i_{qs} \tag{26}$$

$$\varphi_{qr} = (L_{lr} + L_m)i_{dr} + L_m i_{ds}, \tag{27}$$

$$\varphi_{xr} = L_{lr} i_{xr} \tag{28}$$

$$\varphi_{yr} = L_{lr} i_{yr}, \tag{29}$$

$$\varphi_{0r} = L_{lr} i_{0r} \tag{30}$$

Torque:

$$T_e = \frac{5p}{4}\left(\varphi_{ds} i_{qs} - \varphi_{qs} i_{ds}\right). \tag{31}$$

Rotor speed:

$$\omega_r = \int \frac{p}{2J}(T_e - T_L), \tag{32}$$

where $R_s$ represents the stator resistance; $R_r$ represents rotor resistance; $L_l$ represents leakage inductance; $L_m$ represents the maximum mutual inductance of the stator to the rotor; U represents voltage; i represents current; $\varphi$ represents total flux linkages; $p\varphi$ represents $d/dt$; J represents the moment of inertia; p represents number of pole pairs; $T_L$ represents load torque; $T_e$ represents electromechanical torque; $\omega_r$ represents the angular speed of the rotor; $\omega$ represents the angular frequency; $\omega_a$ represents the angular velocity at which the machine equations of any reference frame are transformed [26–28].

We introduced the above model equations using function blocks in the Matlab/Simulink environment version R2019b from MathWorks company. The individual blocks were interconnected using mathematical blocks. An electrical model of the induction machine was created and described by Equations (11–30), and a mechanical model by Equations (31) and (32). The electrical model equation is in a *dq* reference frame. Therefore the transformation (Equation (10)) from *abcde* to *dq* coordinates is used. Subsequently, the inverse transformation was used to obtain quantities in the *abcde* coordinate system. The output of the model are quantities such as currents, fluxes, electromagnetic torque, and rotational speed.

When creating the model, it is necessary to realize that the transformation of the motor voltages from *a, b, c, d, e* to the *dq* plane is created from the voltages of the stator windings. In the star connection, the stator winding voltages are the same as source voltages. Thus, we can use voltages from the source directly for the transformation. However, this does not apply to the pentagon and pentacle connection. Here, the voltages on the stator windings are *1.1756* and *1.902* times higher than source voltages, as shown in Figure 3. Therefore, we have to directly measure the voltage on the stator windings and bring it into the transformation. On the other side, we can multiply source voltages by a constant of *1.1756* for the pentagon and *1.902* for the pentagram.

## 4. Simulation

The simulations given in this chapter were performed for the machine parameters shown in Table 1. The simulation results were subsequently verified by measurements in Section 5. In the simulation verification, we focused on the behavior of torque, power, input power, and losses of a five-phase induction motor in a fault-free state and in fault states, where we simulated the failure of one phase, two adjacent phases, and the failure of two non-adjacent phases. Thus, the five-phase induction motor is investigated in five phases, four phases, and three phases operation.

**Table 1.** Five-phase induction motor parameters.

| Name | Signature | Value | Unit |
|---|---|---|---|
| Stator resistance | $R_s$ | 15.05 | $\Omega$ |
| Rotor resistance | $R_r$ | 5.926 | $\Omega$ |
| Stator inductance | $L_s$ | 0.8714 | H |
| Rotor inductance | $L_r$ | 0.8714 | H |
| Mutual inductance | $L_m$ | 0.85 | H |
| Number of pole pairs | p | 2 | - |
| Moment of inertia | J | 0.007 | $kg \times m^2$ |
| Mechanical power | $P_n$ | 1.1 | kW |
| Source voltages | $U_{a\text{-}e}$ | $5 \times 230$ | V |

The input power was obtained in the simulations as follows. The voltage and current values were measured in simulation using Simulink blocks "Voltage measurement" and "Current measurement." RMS (Root Mean Square) values of voltages and currents were calculated using "RMS value" block to obtain the power of each input phase. Subsequently, the phase shift between voltage and current on a given phase was determined. According to Equation (33), the average value of the input power at each phase was found out. Subsequently, these input powers were added, and we obtained total input power:

$$P_{AV} = U_{rms} \times I_{rms} \times cos\varphi, \tag{33}$$

where:

- $P_{AV}$ is an average value of input power of one phase (W)
- $U_{rms}$ is rms value of input voltage (V)
- $I_{rms}$ is rms value of input current (A)

The mechanical power on the shaft was determined using Equation (4), where we used a mathematical model to determine the torque and angular velocity given by Equations (31) and (32). Then the motor losses were determined as the difference of the mechanical power on the shaft to the input power of the motor.

### 4.1. Simulation Verification of Torque for Fault Conditions

The five-phase induction motor was supplied with a voltage of *5 × 230 V, 50 Hz*, during the simulation in a fault-free state.

Figure 4 is a simulation of a torque characteristic in a star connection of stator windings. Torque characteristics were performed for the fault-free state, phase failure *a*, *ab* and *ac*. The rated motor load is 3.5 Nm. We found out a decrease in the torque for this value of the rated torque in the fault-free state and fault statets. From the simulation of the torque characteristic for connection to the star, we can see that with the loss of one phase, the motor torque at the nominal load will decrease by 38.62%. In the event of a failure of two adjacent phases (*ab*), the torque decrease is 65.28% relative to the fault-free state. And in the event of a failure of two non-adjacent phases (*ac*), this decrease is up to 70.77%.

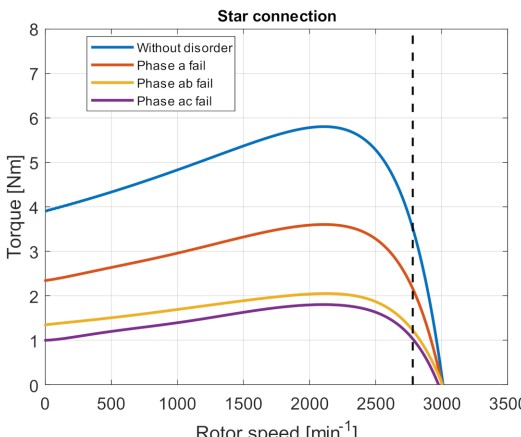

**Figure 4.** Torque characteristics in fault-free state and fault states in a star connection, $U_{IN} = 5 \times 230$ V, 50 Hz.

Figure 5 shows the same simulation of the torque characteristics for the pentagon connection. The supply voltage is the same as for the star connection. It can be seen from Figure 5 that the motor torque has increased too. The percentage decrease in motor torque connected to the pentagon in the event of a single-phase failure is 39.23%. In the event of a failure of two adjacent phases (*ab*), it is 65.20%, and in the event of a failure of two non-adjacent phases (*ac*), it is 73.72%. The percentage decrease is essentially the same/similar, but the resulting torque is *1.37* times greater.

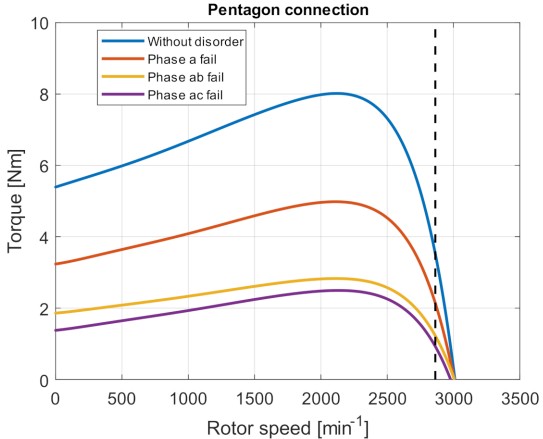

**Figure 5.** Torque characteristics in fault-free state and fault states in pentagon connection, $U_{IN} = 5 \times 230$ V, 50 Hz.

Figure 6 shows a simulation of the torque characteristic under fault conditions for the pentacle connection. We can see from the courses that in the event of one phase failure, there is a decrease of 45.15%, in the event of a failure of two non-adjacent phases, it is 38%, and in the event of a failure of two adjacent phases, there is a decrease of 72.30%. The resulting torque of the machine is *3.62* times greater.

From the waveforms, we can see that the torques, when connected to the pentagon, are *1.369 ($1.17^2$)* times greater and *3.61 ($1.9^2$)* times greater with the pentacle. It means that the five-phase induction motor has the same properties/characteristics when connected to a star as when connected to a pentagon if the supply voltage is reduced by *1.17* and for the pentagram when the supply voltage is reduced by *1.9* times.

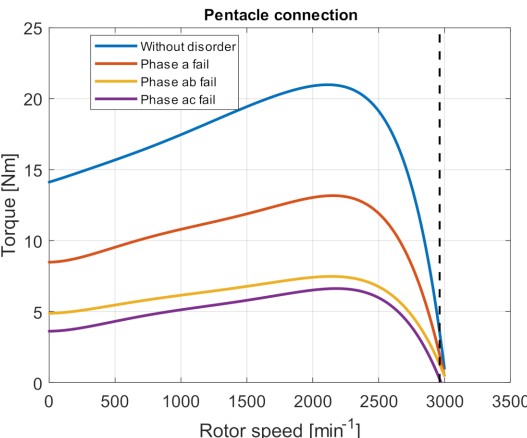

**Figure 6.** Torque characteristics in fault-free state and fault states in pentacle connection, $U_{IN} = 5 \times 230$ V, 50 Hz.

*4.2. Simulation Verification of Input Power and Mechanical Power for Fault Conditions*

In this chapter, the mechanical power on the shaft and the active power taken from the system is simulated. The relations for calculating these powers are given in Section 2, Equations (2) and (4).

Figure 7 shows simulations of a five-phase induction motor mechanical power and power input for all three stator winding connections in a fault-free state. From the waveforms, we see that the mechanical power on the motor shaft is the same for all three connections up to a load of 3 Nm. As the load increases gradually, we see differences between the individual connections. In the pentagon connection (red curve), the power gradually increases while also increasing the motor maximum load torque. With this connection, the maximum torque increases by 39.6%, and the power on the shaft increases by 38.2%, compared to the star connection (blue course). The pentacle connection (yellow waveform) provides an even greater increase in electromagnetic torque and power on the shaft. The increase in torque is up to 270.6%, and the power increased compared to the star connection by up to 261.7%. However, this also increases the power taken from the system, as we can see in Figure 7. However, this will be better considered in the waveforms of losses in Section 4.3.

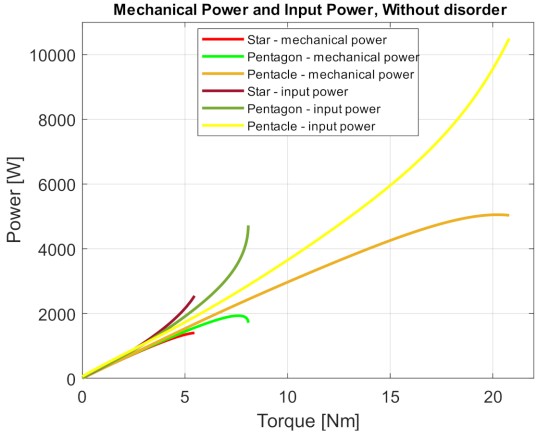

**Figure 7.** Power and power input depending on load torque, fault-free state, $U_{IN} = 5 \times 230$ V, 50 Hz.

Figure 8 shows the dependence of the power and input power on torque, but for the failure state of phase a. The mechanical power is the same for all three connections up to a load torque of 2.2 Nm. The percentage increase in maximum torque and power

during phase *a* failure for connection to the pentagon and pentagram was the same as in the fault-free state.

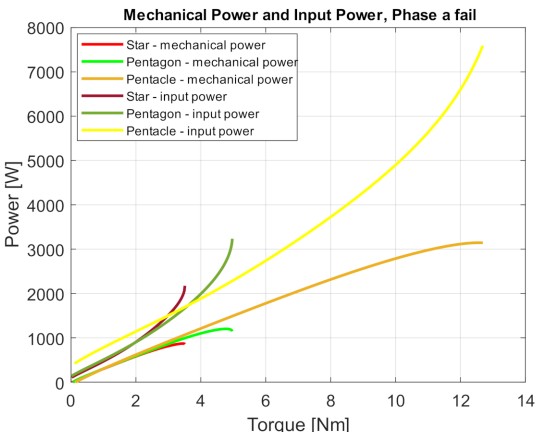

**Figure 8.** Power and power input depending on load torque, phase failure *a*, U$_{IN}$ = 5 × 230 V, 50 Hz.

Figure 9 shows the waveforms of the simulations of mechanical power on the shaft and input power as a function of torque in the failure state of phase failure *ab*. From the waveforms, we can see again that the percentage increase of the maximum load torque and the maximum power on the shaft is the same as in the previous two states (fault-free state and failure of phase *a*).

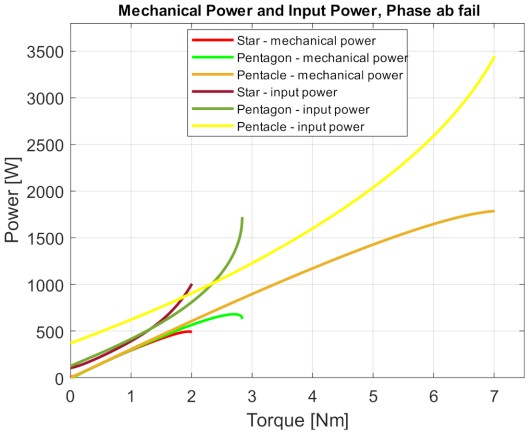

**Figure 9.** Power and power input depending on load torque, phase failure *ab*, U$_{IN}$ = 5 × 230 V, 50 Hz.

Figure 10 shows the waveform of powers simulations for the fault condition of two non-adjacent phases *ac*. Again, we see that the percentage increase in maximum torque and shaft power is the same for each stator windings connection. However, there has been a relatively large increase in input power, which will represent increased losses, as will be discussed in the next chapter.

In terms of the increase in maximum torque and mechanical power on the shaft for the pentagon and pentacle (pentagram) connection compared to the star connection, the increase is the same for all fault states as well as in the fault-free state. Thus, we found that if we compare the individual connections of the star, pentagon, and pentacle to each other, then the increase in power will be the same in all states (without failure, fault in *a*, *ab*, or *ac phases*).

It can be seen from the pictures that in the event of a motor phase failure, the power taken from the source will increase. This phenomenon is most pronounced in the pentagram circuit and also shows an increased consumption in unload operation.

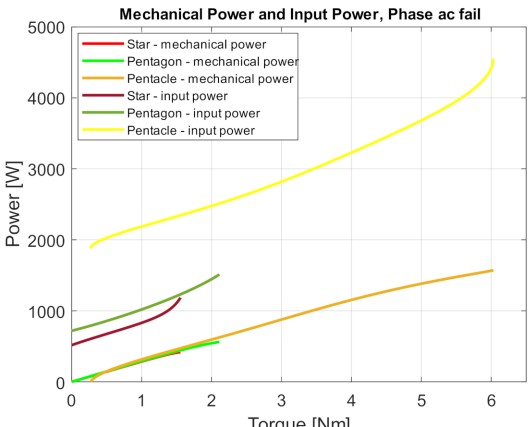

**Figure 10.** Power and power input depending on load torque, phase failure *ac*, $U_{IN} = 5 \times 230$ V, 50 Hz.

In case of failure of one phase, the pentagon connection seems to be the most suitable. The same applies to the failure of two adjacent phases. It follows from the given waveforms that in the event of failure of two non-adjacent phases, operation in any connection is impossible. Even in the event of a failure of two non-adjacent phases, temporary operation of the motor is not possible, as the power loss exceeds the nominal motor power.

Another important comparison is the dependence of power on the torque. The next figures (Figures 11–13) show how the motor mechanical power will be reduced for the star, pentagon, and pentacle connections in fault conditions. Figure 11 shows precisely simulation waveforms in the star connection, from which we can see how the maximum power on the shaft gradually decreased in individual fault conditions. In the event of a phase *a* failure, there was a decrease in mechanical power on the shaft by 37.73%. In the event of *ab* phases failure, the mechanical power is decreased by 64.67%, and in the event of *ac* phases failure, the shaft power is decreased by 71.50%.

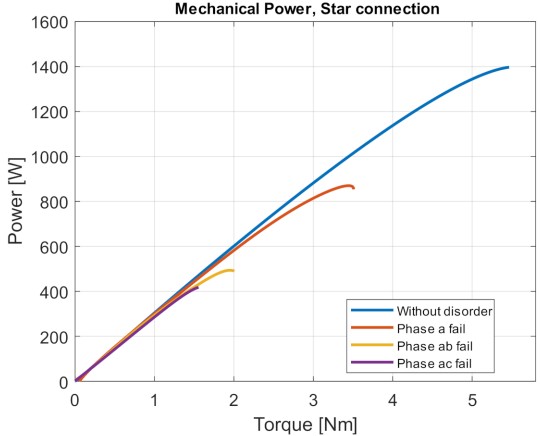

**Figure 11.** Power on the shaft from the load moment, star connection, $U_{IN} = 5 \times 230$ V, 50 Hz.

Figure 12 shows simulated waveforms for the pentagon connection. From the waveforms, we can see that the decrease of the maximum power at all faults compared to the fault-free state is the same as in the star connection.

Figure 13 shows a simulation of the mechanical power of the motor connected to the pentagram. From the waveforms, we can see again that the decrease in power during phase failures is the same as in the previous connections.

It should be noted that when connected to the pentagram, the motor achieves the highest power but at the expense of losses. However, this is uneconomical. Losses are listed in the next subchapter.

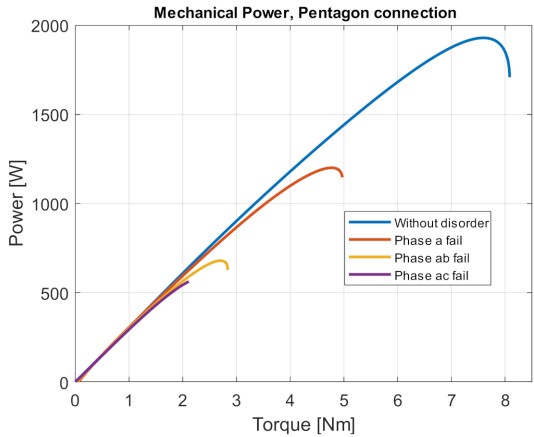

**Figure 12.** Power on the shaft from the load moment, pentagon connection, $U_{IN}$ = 5 × 230 V, 50 Hz.

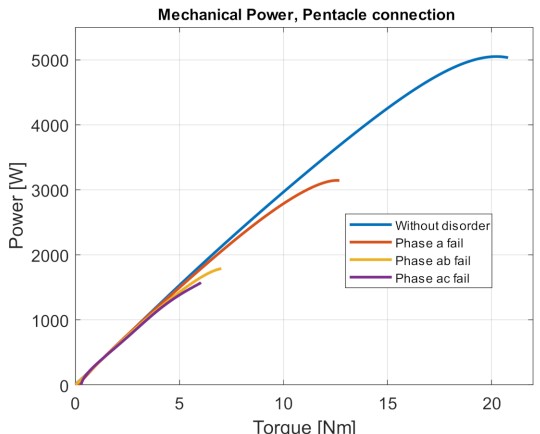

**Figure 13.** Power on the shaft from the load moment, pentacle connection, $U_{IN}$ = 5 × 230 V, 50 Hz.

*4.3. Simulation Verification of Power Losses for Fault Conditions*

This chapter shows the losses of the five-phase induction motor in the fault-free state, i.e., during operation with all five phases, and the losses in the fault states. The following figures show the losses in individual connections. Figure 14 shows the operation of the five-phase motor in a fault-free state. From the graph, we can see that at nominal load, 3.5 Nm, the pentacle connection has the smallest losses. On the other side, the biggest losses are in the star connection. However, in the case of a small load from 0 Nm to 2 Nm, the pentacle connection is characterized by the largest losses shown in Figure 14.

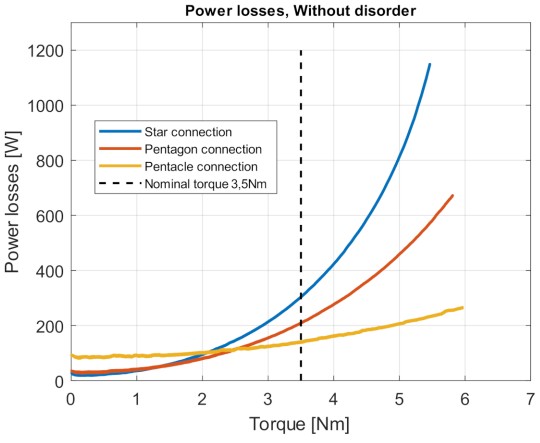

**Figure 14.** Motor losses, fault-free condition, $U_{IN}$ = 5 × 230 V, 50 Hz.

It follows from this figure that it makes sense to run the motor (economic operation) to the point where the losses in the pentagon/pentagram circuit equal the losses in the star circuit at the nominal torque/power. Economical operation is up to about 300 W of power loss, or something more, but only to the point where the motor reaches 80% of the nominal torque.

All operations above 300 W are unsuitable, and motor power should be reduced to minimize this loss. Figure 15 shows that it does not make sense to use pentacle connection even in the event of a single-phase failure, only if we accept larger losses.

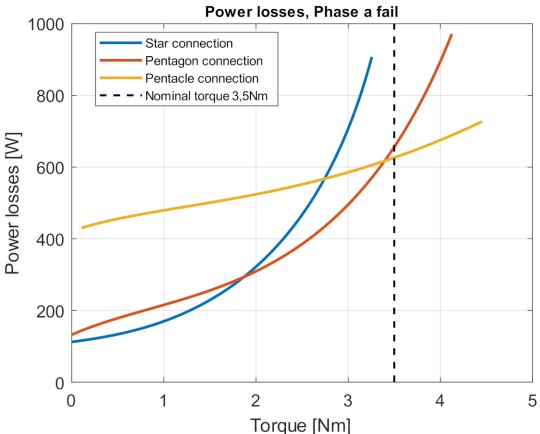

**Figure 15.** Motor losses, phase failure a, $U_{IN} = 5 \times 230$ V, 50 Hz.

Figure 15 shows motor losses in various stator winding connections in a fault condition —failure of one motor phase. We can see from the waveform that with the loss of one motor phase in the pentacle connection, the electric losses increased significantly, and the losses in the star connection and the pentagon are smaller in the first half of the characteristic, but from 1.8 Nm, the losses of the pentagon connection are significantly smaller.

Figure 16 again shows the loss waveforms, but for the failure of two adjacent phases –*ab* phases. We can see from the waveform that the motor does not reach the nominal torque with this connection. The biggest losses are in the pentacle connection. The losses in the star and pentagon connection are smaller like the previous waveforms in Figure 15. There are smaller losses in the star connection in the beginning, and in the case of a load bigger than 1.2 Nm, the smaller losses are in a pentagon connection.

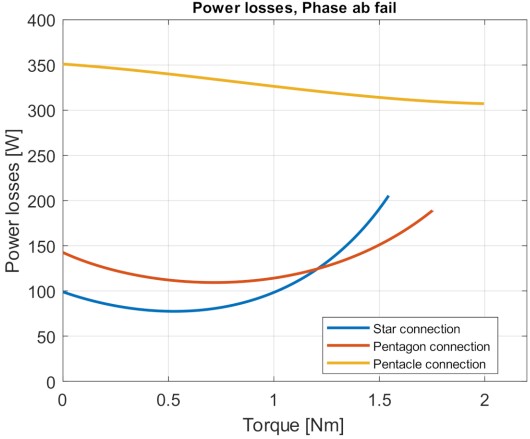

**Figure 16.** Motor losses, phase failure ab, $U_{IN} = 5 \times 230$ V, 50 Hz.

Figure 17 shows simulation waveforms of motor losses when the motor runs with the failure of two non-adjacent phases *ac*. We see from these waveforms that the smallest losses

are in the star connection, and the largest losses are almost threefold when connected to the pentacle.

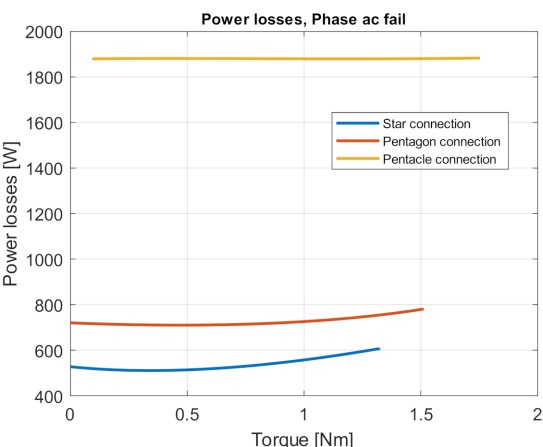

**Figure 17.** Motor losses, phase failure ac, $U_{IN}$ = 5 × 230 V, 50 Hz.

From the simulation waveforms of the five-phase induction motor, we can say that the pentacle connection has the best properties, i.e., the smallest losses in the fault-free state, but in the operation of the motor with a fault (phase failure *a*, *ab* or *ac*), it has the largest losses. Although the connection to the star has the largest losses in the fault-free state, the losses are almost comparable to the pentagon connection in the event of a phase failure. The star connection has better properties in the first half of the characteristic, but the pentagon has smaller losses at higher loads. Therefore, the pentacle connection is inappropriate in this aspect. However, connection in a pentagon has significantly better properties than star connection.

However, in the event of a two-phase failure, the losses are very large, reaching up to 50% of the nominal power. Therefore, operation in this mode is not possible, only in star connection for emergency travel.

### 4.4. Simulation Verification of Power Losses for Various Connections at Constant Magnetic Flux

In this subchapter, the power losses of the five-phase induction motor in the various connections of the stator windings are again presented. However, a constant motor magnetic flux is considered during all simulations.

First, we simulated all three connections of stator windings with the same magnetic flux of motors in fault-free operation. It can be seen in Figures 18–20. These represent the magnetic stator fluxes in the star, pentagon, and pentagram connection. The same magnetic flux was achieved in all three cases by reducing the supply voltage. In the star connection, the supply voltage is $U_{IN}$ = 5 × 230 V, 50 Hz. When connected to a pentagon, $U_{IN}$ = 5 × 196 V, 50 Hz and when connected to a pentagram, $U_{IN}$ = 5 × 121 V, 50 Hz. The blue curve of the circle chart represents a no-load operation. The red curve represents the operation of the motor at a nominal load of 3.5 Nm. Power losses during such operations are listed in Table 2.

Next, Figures 21–23 show the stator magnetic fluxes of the individual stator winding circuits in single-phase failure operation. Again, these are circular diagrams, where the blue waveform is without load, and the red waveform is the nominal load. Power losses are listed in Table 2.

Figure 24 shows stator magnetic fluxes for star connection in operation with a failure of two adjacent phases. Again, this is a circular diagram where the blue curve is unloaded. The red curve represents operation at a load of 2 Nm. Power losses are listed in Table 2. As we can see with the free-fault operation and phase failure *a*, the circular diagram is the same for all three stator winding connections. Therefore, in the case of a two-phase error, we have listed only one connection.

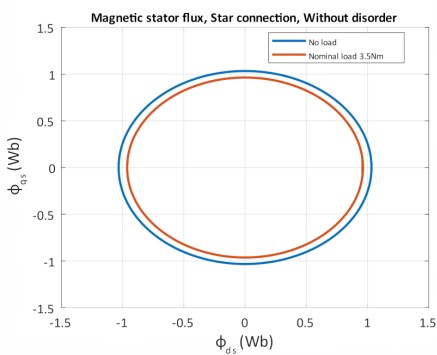

**Figure 18.** Magnetic stator flux, star connection, without disorder.

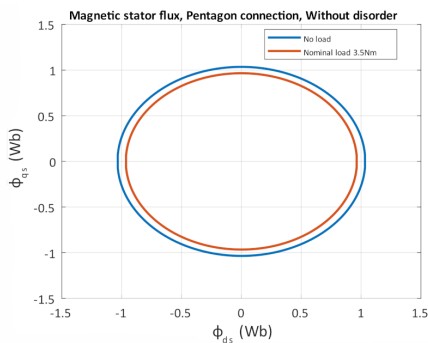

**Figure 19.** Magnetic stator flux, pentagon connection, without disorder.

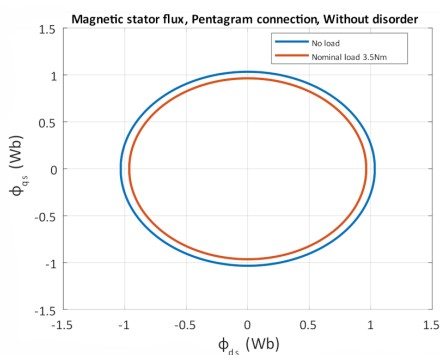

**Figure 20.** Magnetic stator flux, pentagram connection, without disorder.

**Table 2.** Power losses for the same magnetic flux.

| Connection | Signature | No Load | Nominal Load (3.5 Nm) | Load (2 Nm) | Unit |
|---|---|---|---|---|---|
| Star—Without disorder | $P_{loss}$ | 49.60 | 165.50 | - | W |
| Pentagon—Without disorder | $P_{loss}$ | 51 | 164.25 | - | W |
| Pentacle—Without disorder | $P_{loss}$ | 49.85 | 163 | - | W |
| Star—Fault phase a | $P_{loss}$ | 251 | 570.90 | - | W |
| Pentagon—Fault phase a | $P_{loss}$ | 250 | 570.20 | - | W |
| Pentacle—Fault phase a | $P_{loss}$ | 249.40 | 569.80 | - | W |
| Star—Fault phase ab | $P_{loss}$ | 206.80 | - | 217 | W |
| Pentagon—Fault phase ab | $P_{loss}$ | 206.50 | - | 216.50 | W |
| Pentacle—Fault phase ab | $P_{loss}$ | 206.90 | - | 215.60 | W |

Table 2 shows the losses during fault-free operation, in the event of a phase failure, and in the ab phase failure, for all three connections. In all three connections, the magnetic flux of the motor was the same for each condition. We can see from Table 2 that the losses are the same. From this, we can state that if we reduce the supply voltage for the individual

connections of the pentagon and the pentagram so that the magnetic flux of the motor is the same, the power losses of the motor will be almost the same.

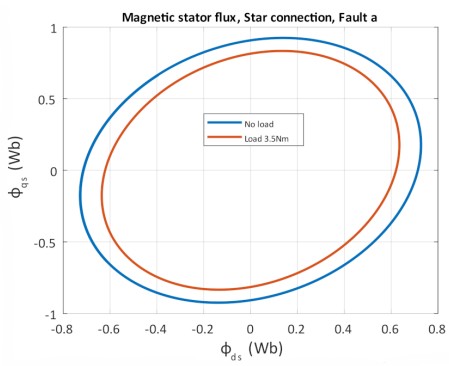

**Figure 21.** Magnetic stator flux, star connection, fault phase a.

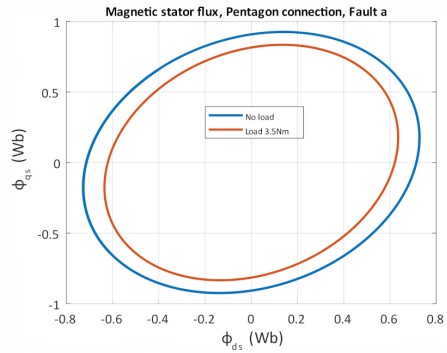

**Figure 22.** Magnetic stator flux, pentagon connection, fault phase a.

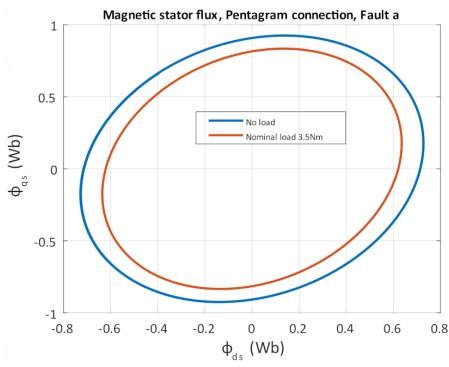

**Figure 23.** Magnetic stator flux, pentagram connection, fault phase a.

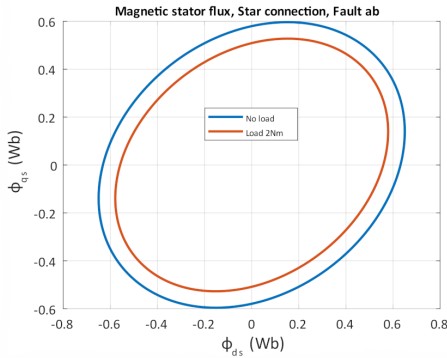

**Figure 24.** Magnetic stator flux, star connection, fault phase ab.

## 5. Measurement and Comparison with Simulation

This chapter presents the measurement of fault conditions on a five-phase induction motor powered by a five-phase VSI and loaded with a dynamometer. The input power of the experimental stage was provided by the voltage source Agilent N8900.Then, the five-phase motor was connected to this input source through a five-phase inverter. That is the difference between measurement and simulation. In the simulation experiment, the five-phase machine was directly connected to the five-phase input source. In the measurement, if this current and voltage of the input source were considered as input power, the inverter losses would be included in the motor losses. Therefore, to obtain the proper value of power losses, RMS values of the phase currents and voltages would be considered as input power and mechanical quantities like angular speed and mechanical torque as output. The electrical quantities like a phase current and phase voltages were measured by the current and voltage probes, and the RMS values were calculated using a four-channel TEKTRONIX TDS 2024B oscilloscope 200MHz, 2GS/s. Therefore, the only current of four phases in a fault-free operation is displayed/measured. It is not a problem because the machine is symmetrical in this state, and the voltages and currents are the same. In a fault operation, only mechanical power was investigated due to unbalanced current and voltages. The torque and angular speed were obtained by the torsion torque sensor and incremental encoder, respectively. According to these values, mechanical power was calculated. Subsequently, the torque-speed characteristics of the motor in various connections were created. The measured phase current at different speeds was compared with a simulation. The fault state was done by the disconnection of the motor phases from the VSI inverter. The results of individual measurements are also performed by simulation and subsequently compared.

In Section 4, where we verified the drive properties by simulation, we were able to do the whole torque characteristic, but in real measurement, only a part of it, due to the heating of the motor under load. Therefore, the torque characteristic is measured only in a certain speed range. Figure 25 shows a circuit diagram of the measurement.

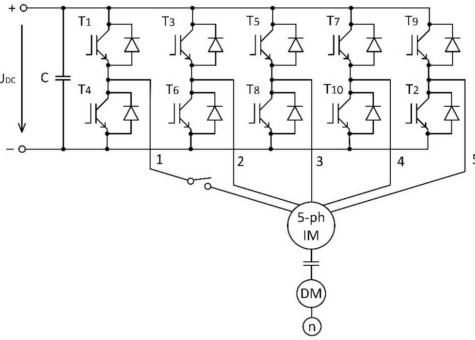

**Figure 25.** Principle circuit diagram for measuring the error states of a five-phase induction motor.

This chapter aims to confirm the correctness of the simulation model with a real motor, so that the results reproduced in Section 4 are the same with a real five-phase induction motor.

### 5.1. Measurement of Torque and Power of 5fIM in Fault-Free State and Case of Failure of One Phase in Star Connection

As mentioned above, the torque characteristic is only measured at certain speeds due to the motor overheating. Figure 26 shows the measurement of the five-phase induction motor torque in a star connection. The red waveform represents the measurement without fault, and the yellow waveform represents the measurement of fault at phase *a*. The motor was powered by a five-phase VSI. DC link voltage was $U_{DC}$ = 500 V. First harmonic voltages $U_{1rms}$ = 170 V. We can see from the graph that in the event of a phase failure, the motor torque decreases to 80% of nominal torque during the normal operation of the machine.

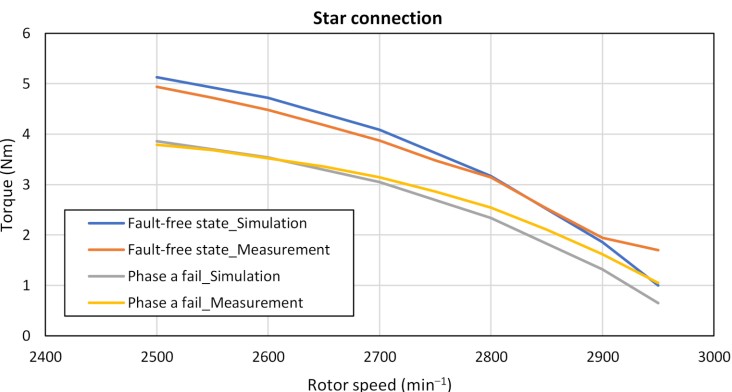

**Figure 26.** Torque measurement and simulation of 5 fIM for fault-free condition and failure of one phase in a star connection.

Figure 26 also shows the simulation curves performed to compare the measurement with the simulation model. The blue waveform represents a fault-free state, and the gray waveform represents the failure of phase *a*. From these two waveforms, we see that the measurement and simulation are almost similar. Thus the rest of the torque characteristic for determining the behavior of the motor is simulated in Section 4.1.

Another measurement to verify the simulation model is the dependence of the motor phase current on the speed in the fault-free state. These waveforms can be seen in Figure 27. The red waveform represents the measured current, and the blue waveform represents the simulated current. We can see that the relative error is 6.1% from these waveforms, which means that the simulation model is almost comparable to a real motor.

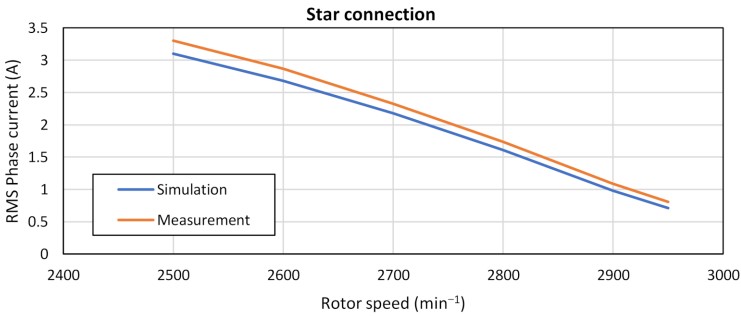

**Figure 27.** Dependence of phase *a* motor current on speed for the fault-free state in the star connection.

In the fault-free state, all currents are the same (symmetrical), with a phase shift of 72°. However, this does not apply if there is a phase failure. In this state, the amplitudes and phase shifts of the individual currents change.

Another part of the investigation, which was verified by measurement, is the measurement of the mechanical power on the shaft depending on the load torque in the fault-free state and a fault state. This dependence in a star connection can be seen in Figure 28. The red curve represents the measurement without fault, and the yellow curve represents the measurement at phase failure. Subsequently, we also verified this measurement by the simulation, shown in Figure 28. The blue curve represents the simulation without fault, and the gray curve represents the simulation at phase failure. By comparing these two graphs, we see that the measurement and simulation match. The maximum relative error is 4%.

### 5.2. Measurement of Torque and Power of 5fIM in Fault-Free State and Case of Failure of One Phase in Pentagon Connection

This subchapter presents measurements on a five-phase induction motor connected to a pentagon. These are the same measurements as in Section 5.1. In Section 4.1, we

found and confirmed that the motor would have the same properties in the pentagon connection as in the star connection if we reduce the input voltage by *1.17* times. Therefore, in this measurement, we reduced the voltage on the DC link of the inverter to the voltage $U_{DC}$ = 420 V, where again the value of the voltage of the first harmonic $U_{1rms}$ = 170 V.

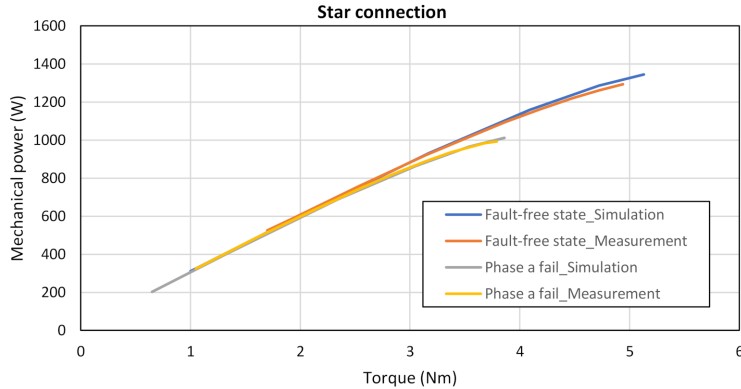

**Figure 28.** The dependence of the mechanical power on the torque for a fault-free state and one phase failure in the star connection.

Figure 29 shows the measurement of the torque characteristic in the fault-free state (gray waveform) and one phase failure (yellow waveform) in the pentagon connection. Figure 29 also shows a simulation of the torque characteristic under the same condition as in measurement (blue waveform for fault-free operation and red waveform for single-phase failure). By comparison, we see that the measurement and simulation are completely identical in the fault-free state and relative error is 5.4%. However, during the phase failure, there was a slight difference in torque during the simulations. The relative error is about 14%.

The same as in star connection, we measured the phase currents in a pentagon connection to verify the comparability of the measurement with the simulations. These currents are shown in Figure 30, where the red curve represents the measurement, and the blue curve represents the simulation. From the waveforms, we see that the relative error is 7%.

The last measurement of this subchapter is the measurement of power on the shaft. It is plotted in Figure 31, where the red waveform is a measurement without failure, and the yellow waveform is a measurement of a single-phase failure. Figure 31 also shows a simulated waveform of the mechanical power, where the blue waveform is fault-free, and the gray waveform has a single-phase failure. Again, we see that the simulation and measurement of waveforms coincide, where the maximum relative error is less than 1%.

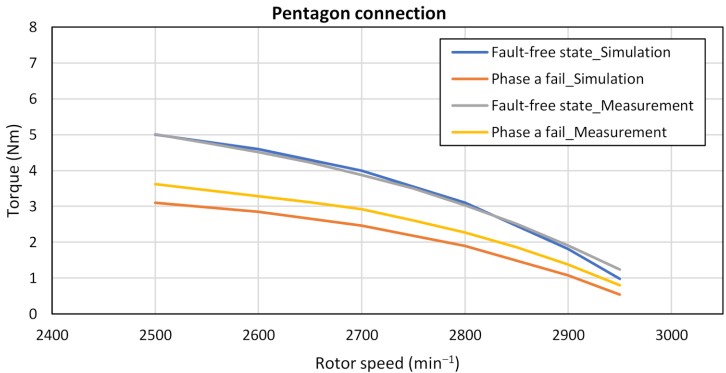

**Figure 29.** Torque measurement and simulation at 5 fIM for fault-free condition and failure of one phase in a pentagon connection.

### 5.3. Measurement of Torque and Power of 5fIM in Fault-Free State and Case of Failure of One Phase in Pentacle Connection

This subchapter represents the measurement of a five-phase machine in pentacle connection. For these measurements, the DC link voltage was $U_{DC}$ = 270 V, and the first harmonic voltage was $U_{1rms}$ = 170 V. Figure 32 shows the torque characteristic in the fault-free state (red curve) and single-phase failure (yellow curve). Figure 32 also shows a simulation where the blue waveform is a fault condition, and the gray waveform is a phase failure. As in the pentagon connection, in the phase failure simulation, the torque is less than in the measurement, and the maximum relative error is 9%.

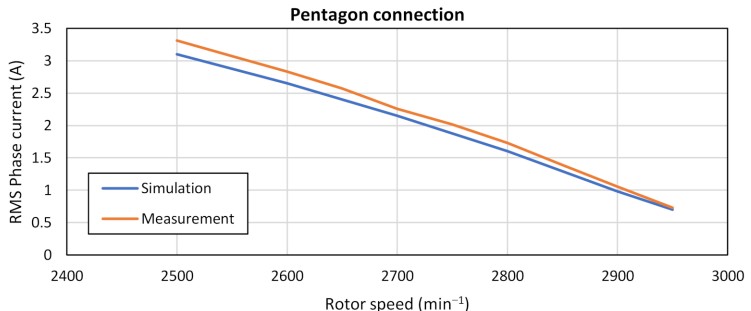

**Figure 30.** Dependence of phase *a* motor current on speed for the fault-free state in pentagon connection.

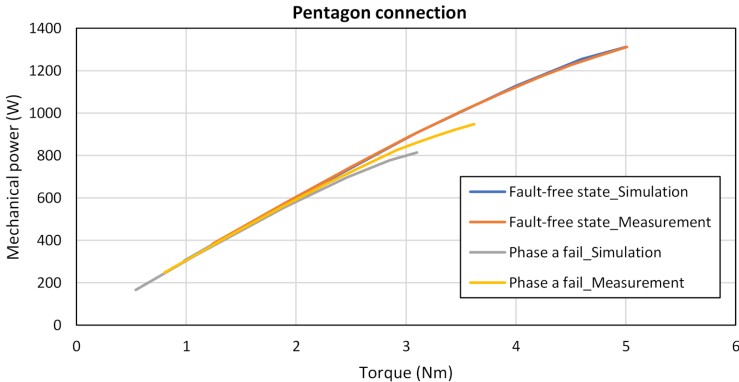

**Figure 31.** The dependence of the mechanical power on the torque for a fault-free state and one phase failure in the pentagon connection.

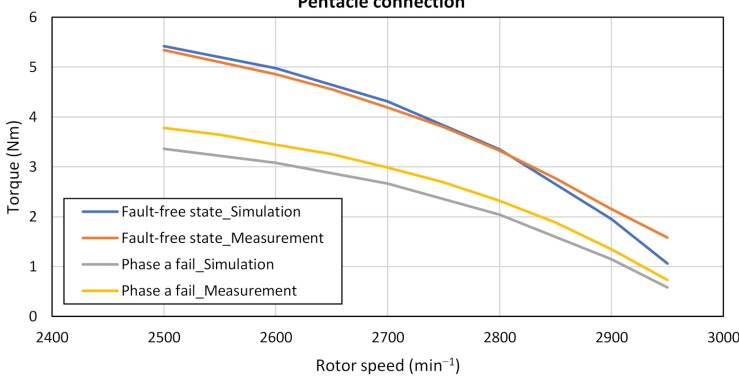

**Figure 32.** Torque measurement and simulation at 5 fIM for fault-free condition and failure of one phase in pentacle connection.

Figure 33 is a waveform from the phase current measurement and simulation in a fault-free state when the stator windings of a five-phase induction motor are connected to

a pentacle. The red curve represents the measurement, and the blue curve represents the simulation. Again, we see that the currents are almost identical, and the relative error is 2.17%.

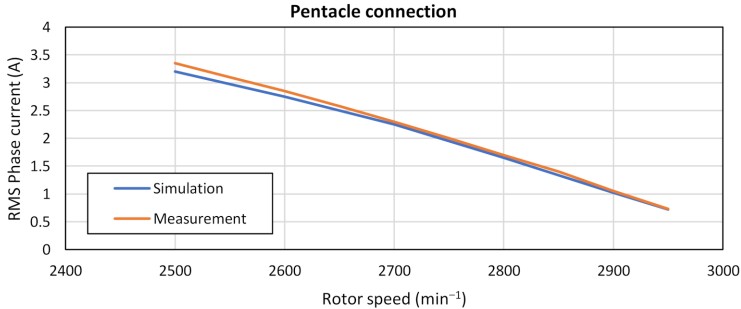

**Figure 33.** Dependence of phase *a* motor current on speed for the fault-free state in pentacle connection.

The last measurement when connected to the pentacle is the measurement of mechanical power. It can be seen in Figure 34, where the red curve represents the measurement in the fault-free state and the yellow in the event of a phase failure *a*. Similarly, a simulation waveform was generated in Figure 34, where the blue curve represents the fault-free simulation and the gray curve the single-phase failure simulation. The relative error is less than 1%.

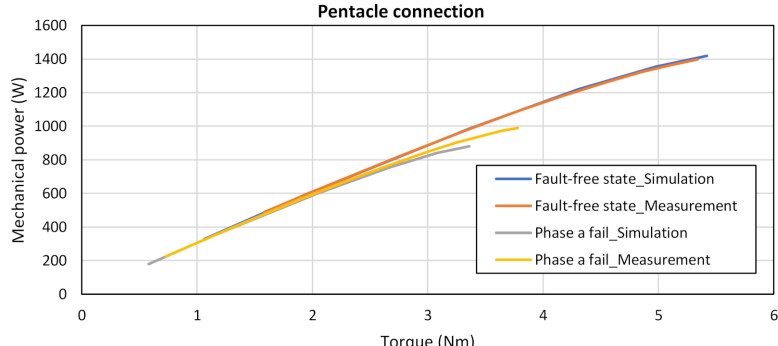

**Figure 34.** The dependence of the mechanical power on the torque for a fault-free state and one phase failure in the pentacle connection.

Figure 35 shows the measurement laboratory, where we can see the measured 5-phase IM, 5-phase VSI, which is built of two three-phase VSI, dynamometer with measurement of speed and torque, notebook for communication between control board from Texas Instrument and user, DC power supply, oscilloscope, and current measurement.

Figures 36–38 shows the waveforms of phase current measurement in individual connections of stator windings. In all cases, the measurement is performed when the engine is idling. Figure 36 shows the phase currents for star connection. The DC link voltage was 500 V, and the first harmonic voltage was 170 V. Figure 37 shows the phase currents for the pentagon connection. The DC link voltage was 420 V, and the first harmonic voltage was 170 V. Figure 31 shows the phase currents for the pentacle connection. The DC link voltage was 270 V, and the first harmonic voltage was 170 V.

The current measurement output has been adjusted. The currents were measured using a current probe with a voltage output. However, it is important that the currents are the same for all three connections because the input voltage was gradually reduced, for the connection to the pentagon by *1.17* times and for the connection to the pentagram (pentacle) by *1.9* times.

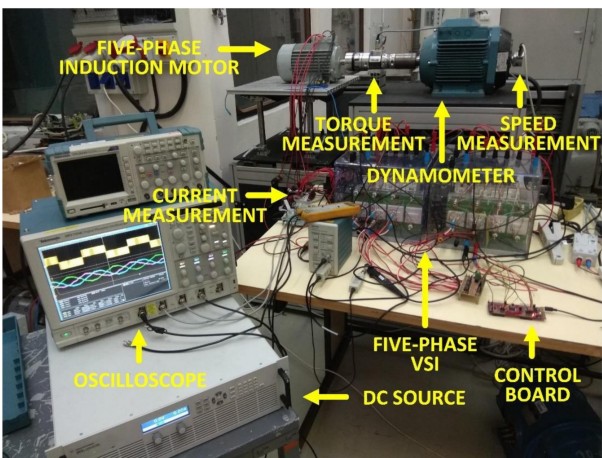

**Figure 35.** Measurement of a five-phase induction motor in fault conditions.

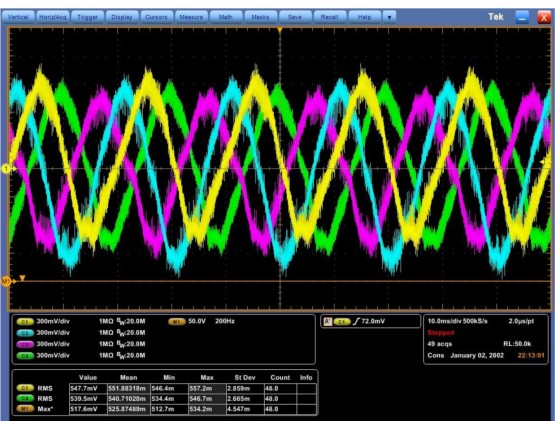

**Figure 36.** Measured motor phase currents in star connection, no load condition.

### 5.4. Conclusion of This Chapter

This chapter shows the 5 phase IM measurement in a fault-free state and failure of one motor phase for the star, pentagon, and pentacle connection. The motor was powered from 5 phase VSI, while the value of the DC link voltage was 500 V, 420 V, and 270 V for the star, pentagon, and pentacle connection, respectively. In all three cases, the value of the first harmonic voltage was 170 V. According to the measured waveforms, and we confirmed that the 5-phase motor has the same properties if we reduce the power supply.

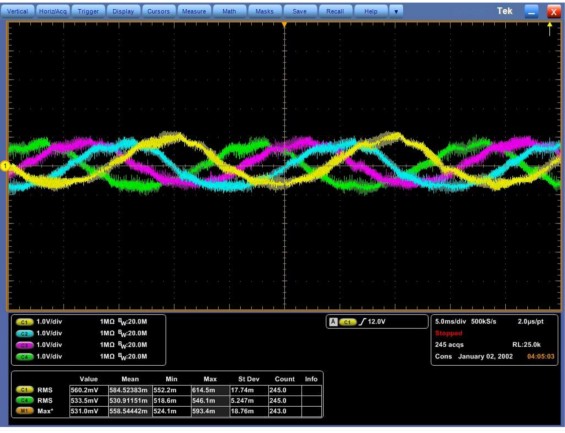

**Figure 37.** Measured motor phase currents in pentagon connection, no load condition.

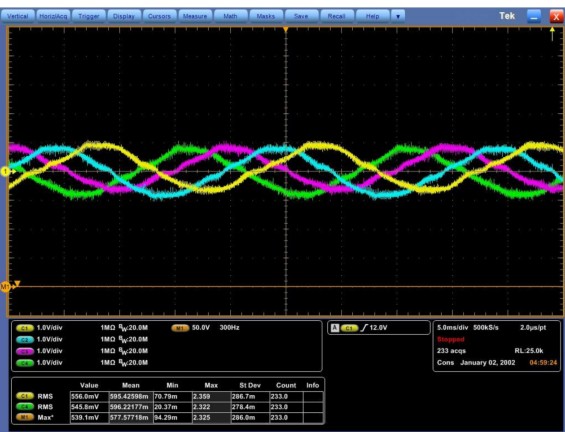

**Figure 38.** Measured motor phase currents in pentacle connection, no load condition.

Other results of this chapter are that we confirmed the correctness of the 5-phase IM model created in the Matlab/Simulink environment. Thus, we confirmed the correctness of the behavioral results of the five-phase induction motor in the fault states listed in Section 4.

## 6. Discussion

The research fundamentals were to find out how the five-phase induction motor will behave in fault states in one phase or two phases failure.

When examining fault conditions, we first focused on the decrease of the torque in phase failure for various connections of stator windings. We found that if we use the same input voltage for all three connections (star, pentagon, pentacle), the torque will increase by *1.37* times in the pentagon connection than the torque in the star connection. When connected to the pentacle, the torque is increased *3.6* times. Furthermore, in the event of a phase failure, the torque decrease at rated load is the same for the star connection and the pentagon connection. It was approximately 40% for the failure of one phase, 65% for the failure of two adjacent phases, and 73% for the failure of two non-adjacent phases. The torque drop at the nominal load in the pentacle connection was very large, as shown in Section 4.1. However, we can say that in terms of the increase in torque at the same input voltage, the pentacle circuit has the best properties.

In Section 4.2., a decrease in the mechanical power and power input from the supply system in fault states for various connections of stator windings was detected. We found that in the fault-free state, the mechanical power on the shaft was the same for all connections up to a load of 3 Nm. In single-phase failure, the mechanical power was the same for all three connections up to a load of 2.2 Nm. When the two phases failed, it was 2.1 Nm.

When connected to the pentagon, the mechanical power will increase by 38.20% compared to the star connection, and when connected to the pentacle, the increase in mechanical power is 261.70%.

Furthermore, we found that in one phase failure, there was a decrease in mechanical power by 37.73%, in a failure of two adjacent phases, the mechanical power decreased by 64.67%, and in a failure of two non-adjacent phases, the mechanical power decreased on the shaft 71.50%, for all connections. Furthermore, this indicates that the motor cannot be operated in a failure of the two phases.

An important part of the research is Section 4.3, which presents the determination of electrical losses for all connections in fault conditions and subsequent mutual comparison. From the waveforms in this chapter, we can see that the connection to the pentacle has the smallest losses and the connection to the star has the largest losses at the nominal load in the fault-free state. However, at loads from 0 Nm to 2 Nm, the losses are greatest in the pentacle connection, as shown in Figure 14. From the waveforms in Figure 15, we can see that the connection in the pentacle has the greatest losses in the event of a phase failure.

We can determine the connection to the pentagon for the best connection, which in the first half of the characteristic has only minimally larger losses than star connection. In the second part of the characteristic, the losses of connection to the pentagon are significantly smaller than in the star connection. The same is true for the failure of two phases.

Section 4.4 investigated the power loss of the induction machine in various connections for a constant flux linkage. Table 2 shows that the power losses are very similar in the case of a constant flux operation.

Section 5 confirmed the simulation results presented in Section 4 by measurements on a real induction motor. Due to the thermal heating of the machine at higher loads, it was not possible to measure the fundamental characteristics because the machine would be destroyed, especially in fault conditions. Therefore, the partially measured characteristics were verified on a simulation model. We found that the simulation characteristics correspond to real measurements on a five-phase induction motor.

## 7. Conclusions

This work presents a comprehensive evaluation of the behavior of a five-phase induction motor in hazardous conditions when one or two phases fail. It should be noted that when using an electric motor to drive an electric vehicle in any configuration, it is necessary to know the behavior of the motor in fault/hazardous conditions. Because in the event of a sudden fault while driving an electric car, stopping and malfunctioning the drive would have fatal consequences. Therefore, this publication aims to find out how the five-phase induction motor will behave in hazardous states in various circuits. We subsequently determined which of the stator windings (star, pentagon, pentacle) has the best properties in terms of power, torque, and energy efficiency, i.e., electrical losses. Thus, which connection is most suitable to use for the design of a five-phase induction motor.

Multi-phase motors are the future of the traction industry due to their several advantages. The properties and suitability for using multi-phase machines in EV and HEV and other industries are listed in Section 1.

Sections 2 and 3 present a five-phase induction motor, a possible connection of stator windings, mechanical construction, and a mathematical model of a five-phase induction motor. Furthermore, the parameters of the induction motor on which the experimental measurements were performed. The relations for the calculation of moments and powers are also given.

Section 4 presents the simulation waveforms of the fault conditions, where the motor torque, mechanical power on the shaft, power consumption from the supply system, and losses on the motor were examined.

Section 5 presents measurements on a real motor, which are then verified on the used 5-phase induction motor model in the Matlab/Simulink environment to verify the accuracy of the motor model.

Section 6 provides a discussion of the identified properties. From the measured and simulated waveforms in the fault states, we found that the most suitable connection of the stator windings of the five-phase induction motor is the pentagon connection. It is the most suitable compromise between the magnitude of power on the shaft and the torque of the motor against the losses in the motor but in the hazardous states. In a fault-free operation, the pentacle connection has the best features.

Here, it is possible to operate 5 phase IM in pentacle connection and in case of motor failure to switch the connection of stator windings to the pentagon using an inverter.

**Author Contributions:** Conceptualization, J.K.; methodology, J.K.; software, S.K.; validation, S.K., J.K. and M.P.; formal analysis, P.R.; investigation, J.K. and S.K.; resources, M.P.; data curation, P.R.; writing—original draft preparation, J.K.; writing—review and editing, J.K. and S.K.; visualization, M.P.; supervision, S.K.; project administration, M.P.; funding acquisition, S.K. and M.P. All authors have read and agreed to the published version of the manuscript.

**Funding:** This research was funded by VEGA 1/0085/21.

**Conflicts of Interest:** The authors declare no conflict of interest.

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
