# Peer review of "A Comprehensive Investigation of the Properties of a Five-Phase Induction Motor Operating in Hazardous States in Various Connections of Stator Windings"

_electronics, doi:10.3390/electronics10050609_

Round 1
Reviewer 1 Report
The paper addresses an absolute interesting topic. The presented calculations and conclusions are valid and of certain interest but for only a completely unsaturated iron circuit. This leads for example to loss evaluation, which is very theoretical. It is clear that with higher voltage the losses are smaller.
Usually an IM is designed with nominal saturation of the iron at no load conditions. This means there is only a small reserve to increase the voltage without changing the number of turns in the winding. Alternatively a huge and unrealistic oversizing of the magnetic circuit is necessary.
I think an additional chapter is necessary to show the influence of the three winding connections at constant flux linkage in the iron circuit. This would give a more realistic insight into the properties of the machine.
Reviewer 2 Report
I have the following technical reservations about the submitted article:
- the simulation part does not describe how the simulations were performed. Only the equations are described, but how the equations were used in the simulation is no longer described.
- it is not information about source of parameters in Tab. 1. Nominal parameters (for individual connections) are not listed for the motor used.
- In equation 3, the parameter Xr20 is incorrectly defined and the parameter Xr1 is missing
- Some graphs look confusing (eg Fig 9)
- There is no better description of the experiment for the possibility of verifying the procedure (eg how the input and output power was measured, etc.)
Formally, I have the following reservations about the article:
- on FIG. 3 shows the connection of the pentagram, but this connection is not shown in FIG. 2.
- on line 140 the word "moment" is used - the author forgot to translate
within the article uses the designation "engine" for the engine (eg line 147) - it is inappropriate to use a different number of decimal places for the numbers being compared
- in some places a decimal point is used, in some places a decimal point
- in some equations the moment is denoted as T, in some places M - it should be unified
- in the descriptions of the designation of equations 11-20 it is not explained what is ρ
- FIG. 22 - the word "satate"
Reviewer 3 Report
the paper is well structured in its parts. the introduction is quite explanatory. The model and simulation phase described in detail. The only thing that can be done is that an "operational" description of the simulations and experimental measurements would be needed, to make everything "repeatable and archiveable". in the judgment the paper is well written and the part joined together satisfactorily. the conclusions are sufficiently argued and the bibliography is confident with the text
Round 2
Reviewer 1 Report
No more objections
Reviewer 2 Report
I have the following technical reservations about the submitted article:
add some references about transformation for five-phase systems
- row 213-214 - "The rms voltage and 213 current value was measured at each input phase." was measured or was calculated?
- use same graphics style for all graphs (Fig 4-24. vs. Fig 26-34)
Unsolved my comments:
There is no better description of the experiment for the possibility of verifying the procedure (eg how the input and output power was measured, etc.) - I mean experiment in chapter 5. There isn't any information about measuring of electrical paramenters.
Some graphs look confusing (eg Fig 9)
